# Evaluation of the Cholesterol-Lowering Mechanism of *Enterococcus faecium* Strain 132 and *Lactobacillus paracasei* Strain 201 in Hypercholesterolemia Rats

**DOI:** 10.3390/nu13061982

**Published:** 2021-06-09

**Authors:** Lingshuang Yang, Xinqiang Xie, Ying Li, Lei Wu, Congcong Fan, Tingting Liang, Yu Xi, Shuanghong Yang, Haixin Li, Jumei Zhang, Yu Ding, Liang Xue, Moutong Chen, Juan Wang, Qingping Wu

**Affiliations:** 1College of Food Science, South China Agricultural University, Guangzhou 510642, China; yangls8272@163.com; 2Guangdong Provincial Key Laboratory of Microbial Safety and Health, State Key Laboratory of Applied Microbiology Southern China, Institute of Microbiology, Guangdong Academy of Sciences, Guangzhou 510070, China; woshixinqiang@126.com (X.X.); liying@gdim.cn (Y.L.); wuleigdim@163.com (L.W.); fancc012@163.com (C.F.); gdim_liangtt@outlook.com (T.L.); xiyu_0604@163.com (Y.X.); 6180112106@stu.jiangnan.edu.cn (S.Y.); 13609778253@163.com (H.L.); zhangjm926@126.com (J.Z.); dingyu@jnu.edu.cn (Y.D.); xueliang@gdim.cn (L.X.); cmtoon@hotmail.com (M.C.)

**Keywords:** cholesterol-lowering, *Enterococcus faecium* strain 132, *Lactobacillus paracasei* strain 201, gut microbiota

## Abstract

Hypercholesterolemia can cause many diseases, but it can effectively regulated by *Lactobacillus*. This study aimed to evaluate the cholesterol-lowering mechanism of *Enterococcus faecium* strain 132 and *Lactobacillus*
*paracasei* strain 201. These results showed that both the strains decreased serum total cholesterol (TC), low-density lipoprotein cholesterol (LDL-C), triglycerides (TG), liver TC and TG and increased fecal TC, TG and total bile acid (TBA) levels. Additionally, both strains also reduced glutamic-pyruvic transaminase (ALT), glutamic oxaloacetic transaminase (AST) and levels of tissue inflammation levels to improve the lipid profile, and they reduced fat accumulation partially by alleviating inflammatory responses. Furthermore, both strains regulated the expression of the *CYP8B1*, *CYP7A1, SREBP-1*, *SCD1* and *LDL-R* gene to promote cholesterol metabolism and reduce TG accumulation. Interventions with both strains also altered the gut microbiota, and decreasing the abundance of Veillonellaceae, Erysipelotrichaceae and *Prevotella*. Furthermore, fecal acetic acid and propionic acid were increased by this intervention. Overall, the results suggested that *E. faecium* strain 132 and *L. paracasei* strain 201 can alleviate hypercholesterolemia in rats and might be applied as a new type of hypercholesterolemia agent in functional foods.

## 1. Introduction

In 2016, approximately 17.6 million deaths were caused by cardiovascular disease globally, which amounted to an increase of 14.5% from 2006 [1]. Individuals with hypercholesterolemia are three times more likely to have a heart attack than those with normal blood lipid levels [2]. Hypercholesterolemia remains one of the largest causes of cardiovascular disease-related death worldwide [3]. The excessive accumulation of triglycerides by hypercholesterolemia in the liver is also considered an important contributor to non-alcoholic fatty liver disease (NAFLD) [4]. People usually adopt a controlled diet to manage their cholesterol levels, but if the levels are too high, the diet is less effective. Statins are the most commonly prescribed cholesterol-lowering drug, and they have a strong efficacy. Lovastatin (Mevacor^®^, Altoprev™) is the most widely used statin used to reduce serum low-density lipoprotein cholesterol (LDL-C) levels, and it consequently reduces the risk of coronary atherosclerosis. [5]. However, several adverse side effects of statins, such as abnormal liver function and muscle pain, among others, have been reported [6]. *Lactobacillus* as non-drug therapies to lower serum cholesterol levels have attracted the attention of scientists. As early as 1963, researchers found that the Samburu tribe in Africa had lower serum cholesterol levels when they ate large amounts of milk fermented with wild lactic acid bacteria or *Bifidobacterium* [7,8]. Recently, several studies have indicated that the probiotic properties of *Lactobacillus* and *Enterococcus* strains, isolated from human or animal guts, effectively lower cholesterol levels [9,10]. In a clinical trial, the probiotic *Lactobacillus reuteri* NCIMB 30,242 helped maintain healthy cholesterol levels in patients with elevated levels [11]. Studies with *Lactobacillus rhamnosus* CGMCC1.3724 [12] and *Bifidobacteriium lactis* HN019 [13] also showed similar positive results. 

The cholesterol-lowering mechanism of lactic acid bacteria is very complex, among which the promotion of cholesterol conversion into bile acid and, thus, the promotion of bile acid excretion is considered to be one of the main mechanisms [14]. Lactic acid bacteria have also been observed to improve liver damage (e.g., inflammation, steatosis, fibrosis, cirrhosis) caused by hypercholesterolemia and decrease obesity [15]. Cholesterol metabolism occurs mainly in the liver, and cholesterol can stimulate macrophages to change from a M2-like phenotype to a pro-inflammatory M1-like phenotype, secreting pro-inflammatory cytokines such as monocyte chemotactic protein-1 (MCP-1), interleukin-1β (IL-1β), interleukin-6 (IL-6) and tumor necrosis factor-α (TNF-α) [16]. Hypercholesterolemia can cause an increase in glutamic-pyruvic transaminase (ALT) and glutamic oxaloacetic transaminase (AST); AST is a mitochondrial damage index, whereas ALT indicates membrane damage. Thus, the changes in these inflammatory factors will affect the hypercholesterolemia and liver function.

*Lactobacillus* can remodel the intestinal microbiota and produce short fatty acids (SCFAs) to intervene with the hypercholesterolemia process [17,18]. Recent research has indicated that the gut microbiota is strongly associated with hyperlipidemia. Patients with hyperlipidemia often exhibit intestinal microbiota disturbances [19]. A person’s diet can shape the intestinal microbiota, which is involved in complex metabolic diseases [20,21,22,23]. It has been reported that diets supplemented with lactic acid bacteria can regulate the intestinal microbiota by inducing SCFA production, and many symbiotic bacteria produce SCFAs in the gut, especially butyrate, acetic acid and propionic acid, which can reduce the risk of gastrointestinal diseases [24,25]. A probiotic intervention influences the intestinal microbiota composition and diversity, which might be effective for the treatment of some diseases. 

Lactic acid bacteria have played an important role in the regulation of hypercholesterolemia. However, the cholesterol-lowering effects and mechanism of *Enterococcus faecium* strain 132 and *Lactobacillus paracasei* strain 201 remain unclear. The aim of this study was to assess two screened human fecal isolates for their cholesterol-lowering capacity in vivo and in vitro and explore the specific mechanism through which these strains protect against liver injury, epididymal fat inflammation, lipid metabolism and variations in the intestinal microbiota associated with hypercholesterolemia in rats.

## 2. Materials and Methods

### 2.1. Cholesterol-Lowering Activity In Vitro

In total, 107 strains of lactic acid bacteria isolated from 29 human fecal samples were selected and stored in the laboratory of the Food Safety and Health Development, Institute of Microbiology, Guangdong Academy of Sciences. The cholesterol-lowering activity of LAB strains was studied in MRS liquid broth containing cholesterol and BA salts (MRS-CHO). The solution (1 g/L) contained cholesterol (0.1 g), tween 80 (1 mL), bile salt (2.0 g), sucrose octaacetate (0.1 g) and glacial acetic acid (1 mL). The components of the solution were dissolved at 60 °C and exposed to ultrasound for 30 min, and then, the prepared MRS liquid medium was quickly added. The MRS-CHO broth was inoculated with a logarithmic-phase bacterial culture (2% *v/v* of OD_600nm_ = 0.5 culture) and incubated at 37 °C for 72 h. After incubation, the bacterial cells were pelleted by centrifugation at 4500× *g* at 4 °C for 10 min. The cholesterol content was determined using a total cholesterol assay kit (Mindray Co., Ltd., Shenzhen, China) in the Mindray BS-480 automatic biochemical analyzer and calculated using Equation (1). *E. faecium* strain 132 and *L. paracasei* strain 201 were screened with a high cholesterol-lowering rate and further investigated.
(1)$$P_{0}\;\left(\%\right)=\left(\frac{D-C}{D}\right)\times 100$$
where P_0_ (%) represents the cholesterol reduction rate, C represents the cholesterol content in the sample and D represents the cholesterol content in the blank control. Each isolate was subjected to three replicates, and the mean ± SD was determined.

### 2.2. Animal and Study Design

In total, 30 specific pathogen-free (SPF) SD male rats (5 weeks of age) were obtained from SMU China (Guangzhou, China). The rats were placed in a controlled environment (temperature 23 ± 3 °C, relative humidity 50–60%, light/dark cycle 12/12 h), and free water and food intake were provided during the experiment. All rats were acclimated for 1 week before starting the experiments. The rats were then randomly divided into five groups, with six rats in each group as follows: (1) control group, fed a Co60 irradiation-maintained diet + 0.9% saline/(mL/100 g/day) intragastrically, (2) HC group, fed a high-cholesterol diet (HCD) + 0.9% saline/(mL/100 g/day) intragastrically, (3) HC-lovastatin group, fed a HCD + 0.1 mg/(mL/100 g/day) lovastatin intragastrically, (4) HC-*E. faecium* strain 132 group, fed 1 × 10^9^ CFU/(mL/100 g/day) *E. faecium* strain 132 + HCD intragastrically and (5) HC-*L. paracasei* strain 201 group, fed 1 × 10^9^ CFU/(mL/100g/day) *L. paracasei* strain 201 + HCD intragastrically. Two strains were cultured in MRS for 18 h, centrifuged at 4000× *g* for 15 min, and the supernatant was removed. The precipitate was washed twice with 0.9% saline and resuspended in 0.9% saline with an OD_600_ = 1.0 (10^9^ CFU/mL). The Co60 irradiation-maintained diet included crude protein ≥180 g/kg, crude fat ≥ 40 g/kg, crude fibre ≥ 50 g/kg, crude ash ≤ 80 g/kg, calcium 10~18 g/kg, phosphorus 6~12 g/kg, Lys ≥ 8.2 g/kg and Met +Cys ≥ 5.3 g/kg. The HCD included the Co60 irradiation-maintained diet, 10% lard, 1% cholesterol and 0.2% BA salt [26]. The weights of the rats were measured weekly for 6 weeks. The experimental design was approved by the Laboratory Animal Management and Ethics Committee of Guangdong Institute of Microbiology (GT-IACUC202005081) and followed the standard guidelines for maintenance. After 42 days of feeding, 30 mg/kg Zoletil 50 (Virbac Co., Ltd., Carros, France) was used for anaesthetization, and blood was collected from the heart; the rats were sacrificed, and the heart, liver, spleen, kidney, epididymal fat, ileum and colon were immediately removed and washed. The heart, liver, spleen, kidney and epididymal fat were weighed, frozen with dry ice and stored at −80 °C.

### 2.3. Serum Biochemical Index Analysis

Blood from the rat heart, collected after a 16-hour period of fasting, was collected for serum biochemical analysis. The blood was centrifuged at 2500× *g* for 10 min at 4 °C, and the serum was collected. Low-density lipoprotein cholesterol (LDL-C), serum total cholesterol (TC), total bile acid (TBA), triglyceride (TG), high-density lipoprotein cholesterol (HDL-C) cholesterol and AST and ALT levels were measured using the BS-480 Reagents on a Mindray BS-480 fully automatic biochemical analyzer (Mindray Co., Ltd., Shenzhen, China) [27].

### 2.4. Liver and Faeces TC, TBA and TG Contents

Lipids were extracted from the separated liver and feces by the Lepercq method [28]. The extracted lipids were used for the determination of TC, TBA and TG levels included in 2.3. The content of cholic acid in feces was determined as described by Kajiura [29]. TC, TBA and TG were measured as in Section with 2.3.

### 2.5. Determination of Related Inflammatory Factors in Liver and Epididymal Fat

Tissue (50 ± 0.25 mg) was dissolved in 500 μL 1× protease inhibitor (Beyotime Biotechnology, Shanghai, China), which was grinded in a grinder (65 Hz, 45 s, −20 °C). The supernatant was collected after centrifugation at 4 °C at 13,000× *g* for 5 min. The inflammatory cytokines MCP-1, IL-1β, IL-6 and TNF-α in liver and epididymal fat were determined using commercially available ELISA kits (Solarbio life sciences Co., Ltd., Beijing, China) according to the manufacturers’ instructions. Total protein content in the liver and epididymis fat was measured with a bicinchoninic acid assay (BCA) protein assay kit (Sangon Biotech Co., Ltd., Shanghai, China). The levels were calculated by a method using Equation (2):(2)$$O\;\left(\frac{\mathrm{pg}}{\mathrm{mg}}\right)=\frac{p}{Q}$$
where O represents the relative values of inflammatory factors and P and Q represent the values of inflammatory factors and total protein content, respectively.

### 2.6. Histopathological Examination

The epididymal fat and liver were rinsed with PBS and fixed in 10% buffered formalin for 24 h. After fixation, the tissues were dehydrated using an alcohol gradient, paraffin-embedded and sliced with a cut angle of 10°. The thickness of liver slices were 3 μm and that of fat slices were 5 μm; tissues were then dewaxed with xylene and stained with haematoxylin, and the ScopeImage 9.0 system was used at a magnification of 200× for observation [30]. Steatosis and hepatocyte inflammation were scored based on non-alcoholic fatty liver disease (NAFS) scores [31].

### 2.7. Real-Time Fluorescence Quantitative PCR (RT-qPCR)

The mRNA expression levels were measured using a LightCycler^®^ 480 SYBR Green I Master system (Roche, Basel, Switzerland). The total RNA was extracted from liver tissue according to the instructions of the RNA Easy Fast Animal Tissue/Cell total RNA extraction kit (DP451, Tiangen Biotech Co., Ltd., Beijing, China), and the RNA concentration was determined with a Nanodrop 2000 UV-vis spectrophotometer (Thermo Fisher Scientific, Waltham, MA, USA). Next, 500 ng RNA was taken and reverse-transcribed using a Fast-King kit (KR116, Tiangen Biotech Co., Ltd., Beijing, China). The specific primer pairs are shown in Table 1 and were designed using Primer Premier 6.0. The 2^−ΔΔCt^ method was used to measure the relative transcript mRNA levels of the hepatic fat metabolism key regulators, such as sterol regulatory element-binding protein 1c (*SREBP-1c*), nuclear fanitol receptor (*FXR*), sterol regulatory element-binding protein 2 (*SREBP-2*), 3-hydroxy-3-methyl-glutaryl-coenzyme A reductase (*HMGCR*), peroxisome proliferator-activated receptor α (*PPARα*), stearyl CoA desaturase 1 (*SCD1*), low density lipoprotein receptor (*LDL-R*), cholesterol 7-α-hydroxylase (*CYP7A1*) and cholesterol 12-α-hydroxylase (*CYP8B1*).

### 2.8. Intestinal Microbial Diversity 

Genomic DNA was extracted from fecal samples of rats treated for 6 weeks (see Section 2.4) using an M6 HiPure Stool DNA Kit (Megan, Guangzhou, China). The concentration and quality of the extracted DNA were measured using a Thermo Scientific Qubit 3.0 (Thermo Scientific Co., Ltd., Waltham, MA, USA). The 16S rRNA gene sequences were amplified with primers specific of the V4-V5 region (515F: GTGCCAGCMGCCGCGG, 907R: CCGTCAATTCMTTTRAGTTT). The Illumina MiSeq platform was used for sequencing, with a depth of 2 × 250 bp. In brief, the paired-end reads were joined after inverse multiplexing. The FASTA quality file and the metadata file representing the barcode sequence of each sample were used as input, and the samples were segmented according to the barcode. The minimum count per sample was 3383, and the average count per sample was 6405. Quality control, rarefaction curve drawing and diversity analysis were performed using the open-source platform QIIME2. The DADA2 method was used for noise removal and quality control, and rooted-trees and feature tables (ASV) were created.

### 2.9. Determination of Short Chain Fatty Acids (SCFAs)

First, fecal samples were homogenized in 0.01% vitriol and centrifuged at 13,000× *g* for 25 min. The supernatant was collected, and volatile free acids were used as an external standard. The SCFAs of the supernatant were used with an Agilent 7693A gas chromatograph (Agilent Technologies, Santa Clara, CA, USA) under the following conditions: 7.2452 psi pressure, 20 mL/min desolvation gas flow, 3 mL/min cone gas flow, 170 °C oven temperature and 250 °C for the flame ionization detector and injection port temperature. Nitrogen was used as the carrier gas. The retention times and peak areas of samples were confirmed using a standard mixture.

### 2.10. Statistical Analysis

The results were reported as the mean ± SD of experiments, which were performed in triplicates. Graph Pad 8.4.3 software was used to calculate the mean and SD values. The statistical significance of differences among groups was calculated with one-way analysis of variance (ANOVA) and multiple comparisons with Tukey’s test. SPSS 25.0 and Graph Pad 8.4.3 software were used for drawing and data analysis, respectively. The Kruskal–Wallis test was used to analyze the alpha diversity index, and the PERMANOVA test was used for beta diversity index analysis. Linear discriminant analysis (LDA) effect size was used for the detection of biomarkers; the false discovery rate (FDR) was used to adjust for microbiota analyses. *p-*values < 0.05 were considered statistically significant.

## 3. Results

### 3.1. Cholesterol-Lowering Ability of Selected Strains

The cholesterol-lowering ability of the isolated strains was screened, and two strains with good ability were found in vitro (Appendix A): the cholesterol-lowering rates were 23.62 ± 6.73% (1302-1) and 25.36 ± 0.62% (2060-11). A BLAST search of the 16S ribosomal RNA sequence (Bacteria and Archaea) database indicated that the 16S rRNA gene sequence of 1302-1 (1422 bp) was 99.8% (difference of 2/1422 bp) similar with that of the *E. faecium* strain DSM 20477. Further, 2060-11 (1430 bp) showed 99.9% (difference of 1/1430 bp) similarity with the *L. paracasei* strain JCM 8130. Therefore, 1302-1 and 2060-11 were named as *E. faecium* strain 132 and *L. paracasei* strain 201, respectively. 

### 3.2. Effects of Screened Strains on Serum Parameters in of Hypercholesterolemic Rats

Body weight gain was significantly different between the HC rat group and other treatment groups (*p* < 0.05; Table 2), and weekly weight gain is shown in Appendix A. The liver and epididymal fat index were significantly higher in the HC group than in the other groups (*p* < 0.05; Table 2). There was no significant difference in other organ weights in each group. Serum TC, TG, LDL-C and TBA values differed significantly between the HC and control groups (*p* < 0.05; Table 2). Compared to those in the HC group (TG: 0.6 ± 0.30 mmol L^−1^, LDL-C: 0.93 ± 0.35 mmol L^−1^), serum TG and LDL-C levels were significantly decreased in the HC*-E. faecium* strain 132 (TG: 0.38 ± 0.08 mmol L^−1^, LDL-C: 0.692 ± 0.12 mmol L^−1^) and *HC-L. paracasei* strain 201 (TG: 0.39 ± 0.03 mmol L^−1^, LDL-C: 0.61 ± 0.06 mmol L^−1^) groups (*p* < 0.05). Serum TC levels were decreased, but serum HDL-C and TBA levels were not affected in the treatment groups, compared to levels in the HC group (Table 2).

### 3.3. Liver and Faecal TC, TBA and TG Levels in Hypercholesterolemic Rats

The liver TC and TG values in the HC group (TC: 9.98 ± 2.35 mmol L^−1^) were significantly higher than those in the control group (TC: 2.24 ± 0.38 mmol L^−1^; *p* < 0.05). Liver TC levels in the HC-*E. faecium* strain 132 (6.12 ± 0.93 mmol L^−1^) group were significantly decreased compared to those in the HC group (*p* < 0.05), and the values were similar to those in the control group (Figure 1A). Liver TBA levels increased but liver TG levels decreased in the other groups compared to those in the HC group (Figure 1B,C). Fecal TC increased in the HC-*E. faecium* strain 132 group compared with that in the HC group (*p* < 0.05; Figure 1D). A significant increase in fecal TBA content was observed in the HC-lovastatin (452.28 ± 42.75 μmol L^−1^), HC-*E. faecium* strain 132 (472.71 ± 45.48 μmol L^−1^) and HC-*L. paracasei* strain 201 (466.51 ± 31.06 μmol L^−1^) groups compared to that in the HC group (401.93 ± 49.86 μmol L^−1^; *p* < 0.05; Figure 1E). Fecal TG was similar among all rats (Figure 1F).

### 3.4. Screening Strains Improved Liver Injury in Hypercholesterolemia Rats

To study whether the screened strains could improve liver injury, in this study, we measured the levels of ALT and AST in serum, determined the content of inflammatory cytokines in liver cells by ELISA and analyzed morphology based on HE staining in the liver. Compared to those in the HC group (ALT: 69.67± 13.70 U/L, AST: 158.68 ± 71.41 U/L), ALT and AST levels were significantly decreased in the HC*-E. faecium* strain 132 (ALT: 42.20 ± 4.87 mmol L^−1^, AST: 100.32 ± 16.29 mmol L^−1^) and *HC-L. paracasei* strain 201 groups (ALT: 45.81 ± 6.53 mmol L^−1^, AST: 97.87 ± 8.27 mmol L^−1^; *p* < 0.05; Figure 2A,B).

Inflammatory factors are associated with many diseases and mainly cause inflammatory changes in the liver with hyperlipidemia. The MCP-1, IL-1β, IL-6 and TNF-α contents in the liver are shown in Figure 2. Compared with levels in the HC group, there was significant inhibition in the HC-lovastatin group (*p* < 0.05; Figure 2C). At the same time, liver IL-1β significantly decreased after treatment with *E. faecium* strain 132 and *L. paracasei* strain 201 compared to levels in the HC group (*p* < 0.05; Figure 2D). Furthermore, liver IL-6 was significantly decreased in the HC-lovastatin group, and TNF-α content was only significantly decreased in the HC-lovastatin and HC-*E. faecium* strain 132 groups (*p* < 0.05; Figure 2E,F). HE-stained liver sections of HCD rats showed intracellular vacuolation and significant lipid accumulation. Compared to that in the HC group, the NAFS score showed that the degrees of steatosis and hepatocyte inflammation were appreciably reduced in HC-lovastatin and HC-*E. faecium* strain 132 groups (*p* < 0.05; Figure 2G,H). 

### 3.5. Screened Strains Improved the Inflammation of Epididymis Fat and Fatty Hypertrophy in Hypercholesterolemic Rats

Epididymal adipose levels of MCP-1 and IL-6 showed significant decreases in the treatment groups compared with those in the HC group (*p* < 0.05; Figure 3A,C). However, there was no significant difference in the content of IL-1β and TNF-α in epididymal adipose among groups (Figure 3B,D). Epididymal fat was observed by HE staining. Compare to that in the HC group (134 ± 8), the number of adipocytes in the same visual field was significantly increased in the control (176 ± 5), HC-lovastatin (173 ± 3), HC*-E. faecium* strain 132 (207 ± 6) and HC*-L. paracasei* strain 201 (190 ± 3) groups (*p* < 0.05), and the diameter of adipocytes was significantly reduced (Figure 3E–G).

### 3.6. mRNA Expression Levels of Genes Related to Lipid Metabolism

To explore the effects of treatment with *E. faecium* strain 132 and *L. paracasei* strain 201 on lipid metabolism in rats, we mainly analyzed the transcript mRNA levels of the key hepatic fat metabolism genes *CYP7A1*, *CYP8B1*, *LDL-R*, *PPARα*, *SREBP-1*, *SREBP-2*, *FXR* and *HMGR*. The relevant data are shown in Figure 4. Data for the mRNA level of *SREBP-2*, *FXR* and *HMGR* showed no significant difference among the groups. Compared with those in the HC group, the expression levels of *CYP8B1* and *LDL-R* were significantly increased, whereas *SREBP-1* was significantly decreased in the HC-*E. faecium* strain 132 group (*p* < 0.05). The expression of *CYP7A1* was also significantly increased (*p* < 0.05), whereas the expression of *SREBP-1* and *SCD1* was significantly decreased (*p* < 0.05) in the HC- *L. paracasei* strain 201 group. Furthermore, the expression of *CYP7A1*, *LDL-R* and *PPARα* was significantly increased in the HC-lovastatin group (*p* < 0.05), whereas the expression of *SREBP-1* was significantly decreased (*p* < 0.05). *E. faecium* strain 132 and *L. paracasei* likely regulated those genes to improve cholesterol metabolism and reduce fat accumulation in the liver.

### 3.7. Gut Microbiota Modulation by Selected Strains

When the rarefaction curve tends to be flat, this indicates that the amount of sequencing data is reasonable. As shown in Figure 5A, when the sequencing depth was deeper, the curve tended to flatten, indicating that the sequencing depth was reasonable. Similarly, with deeper sequencing, the Shannon index tended to be flat (Figure 5B). In our study, we used the Chao1, Shannon and Simpson indexes to analyze the alpha diversity (richness and evenness), and the result showed that the three indexes in the HC-lovastatin and HC*-E. faecium* strain 132 groups were significantly improved compared with those in the HC group, but no significant difference was detected in the HC*-L. paracasei* strain 201 group (*p* = 0.60, *p* = 0.37, *p* = 0.37; Figure 5C–E), indicating that the HC-lovastatin and HC*-E. faecium* strain 132 groups showed high richness and diversity. The PCoA plot (unweighted UniFrac distance) showed that the gut microbiomes of different groups were well separated from each other, and each group was clustered in a different circle (Figure 5F). Next, PERMANOVA multi-dimensional statistical analysis was performed; the control group (*p* = 0.004), HC-lovastatin group (*p* = 0.003), HC-*E. faecium* strain 132 group (*p* = 0.014) and HC*-L. paracasei* strain 201 group (*p* = 0.035) were statistically different compared to the HC group.

At the phylum level, Firmicutes and Bacteroidetes comprised most of the microbiome of each treated rat group. The ratios of Firmicutes/Bacteroidetes were 1.16, 1.63, 2.13, 1.98 and 2.25 in the control, HC, HC-lovastatin, HC*-E. faecium* strain 132 and HC*-L. paracasei* strain 201 groups, respectively (Figure 5G). The relative abundance of Proteobacteria showed no evident differences between the groups, but the proportion of Tenericutes was significantly higher in the control group than in the HCD group, and the relative abundance of Actinobacteria showed the opposite trend (Figure 5G). At the class level, the proportions of Bacilli and Bacteroidia were significantly higher in the control group than in the HCD groups, but the abundance of Clostridia was higher in the HCD groups (Appendix A). At the order level, the bacterial abundance was similar to that at the class level (Appendix A). At the family level, after intervention with HC-lovastatin, HC*-L. paracasei* strain 201 or HC*-E. faecium* strain 132, the abundances of Veillonellaceae and Prevotellaceae were significantly lower than those in the HC group (*p* < 0.05), and bacteria belonging to the families Bacteroidaceae, Peptostreptococcaceae, Lactobacillaceae, Paraprevotellaceae, Coriobacteriaceae and Erysipelotrichaceae were also less abundant than those in that the HC group (Figure 5H and Appendix A). However, the abundances of Ruminococcaceae, S24_7, Lachnospiraceae, Spirochaetaceae and Corynebacteriaceae in the HC-lovastatin, HC*-L. paracasei* strain 201 or HC*-E. faecium* strain 132 groups were higher than those in the HC group. A high abundance of Veillonellaceae and Erysipelotrichaceae was observed in the fecal samples of the HC group (Figure 6A,B, *p* < 0.05). At the genus level, the heat map indicated that *CF231*, which belongs to the family Paraprevotellaceae, was significantly enhanced in the control group compared to levels the other groups. *Prevotella*, *Succinispira*, *Bacteroides*, *Ruminococcus*, *Paraprevotella*, *Collinsella* and *Eubacterium* were decreased in the HC-lovastatin, HC*-E. faecium* strain 132 and *HC-L. paracasei* strain 201 groups compared to abundances in the HC group. Conversely, there was a lower abundance of *Corynebacterium*, *Lactobacillus*, *Clostridium*, *Butyricicoccus* and *Lactonifactor* in the HC group compared to that in the HC-lovastatin, HC*-E. faecium* strain 132 and HC*-L. paracasei* strain 201 groups ( Figure 5I and Appendix A). Differences in the abundance of *Prevotella* and *Butyricicoccus* were statistically significant (Figure 6C,D, *p* < 0.05). 

After FDR-adjustment, the abundances of Veillonellaceae, *Eubacterium*, Coriobacteriaceae, Erysipelotrichaceae, Prevotellaceae and *Ruminococcus* in the HC group were noticeably increased compared to their abundances in other groups (Figure 7). The LDA scores of *CF231* and Paraprevotellaceae were significantly increased in the control group. The HC-*L. paracasei* strain 201 group showed a significantly higher proportion of Ruminococcaceae and Lachnospiraceae, and the abundance of *Sporobacter* was significantly increased in the HC*-E. faecium* strain 132 group (Figure 7). 

### 3.8. SCFA Content in Faeces in Hypercholesterolaemic Rats

To investigate the amount of SCFAs produced by bacteria in the gut, we used gas chromatography to determine fecal contents in this study. First, we measured total contents of acetic acid, propionic acid, isobutyric acid, butyric acid, isovaleric acid and valeric acid, and the results are shown in Table 3. The levels of isobutyric acid, butyric acid, isovaleric acid and valeric acid were not significantly different among the groups. However, the acetic acid and propionic acid levels were significantly increased in the HC-lovastatin and HC*-E. faecium* strain 132 groups, whereas the acetic acid level was not significantly different in the HC-*L. paracasei* strain 201 group compared to that in the HC group. These results suggested that acetic acid and propionic acid levels might be the differential metabolites.

## 4. Discussion

Lactic acid bacteria have been widely used to regulate lipid metabolism, inflammation levels, obesity and gut microbiota [32], which have aroused considerable attention due to fewer side effects and free availability. In this study, we investigated the cholesterol-lowering effects and the potential mechanism of *E. faecium* strain 132 and *L. paracasei* 201 in hypercholesteremia rats induced by high cholesterol diets.

The result showed that *E. faecium* strain 132 displayed lowered serum TC levels, LDL-C, TG and liver TC levels and significantly higher faecal TC and TBA levels. In our cognitive scale, we know that CYP7A1 and CYP8B1 act as a rate-limiting enzymes in the classical pathway of bile acid anabolism [12]. Our results showed that the *E. faecium* strain 132 and *L. paracasei* strain 201 might up-regulate the expression of *CYP8B1* and *CYP7A1* genes, respectively, to promote the reduction in cholesterol levels in rats. Subsequently, high LDL-C levels reportedly correlate with a risk of coronary artery disease [33], and liver LDL-R receptors promote LDL transport. The expression of the *LDL-R* gene was increased after treatment with lovastatin and *E. faecium* strain 132, suggesting that this can accelerate LDL transport and reduce LDL-C levels in the body. However, because SREBP-1 can induce the expression of adipogenic genes and mediate the formation of TG and lipid accumulation, the SCD1 is a liposynthetase [34]. In this study, we discovered that the *E. faecium* strain 132 and *L. paracasei* strain 201 might reduce the expression of the *SREBP-1* gene and that the *L. paracasei* strain 201 also reduces the expression of the *SCD1* gene to suppress TG accumulation. These results suggest that *E. faecium* strain 132 and *L. paracasei* strain 201 might regulate the expression of genes related to lipid metabolism to promote cholesterol excretion and inhibit the accumulation of fat in the liver.

The liver is the main site of lipid metabolism, which is susceptible and vulnerable. In the process of hepatotoxicity, damaged hepatocytes release liver-specific enzymes, such as ALT and AST, into the bloodstream, causing an elevation of both in the serum [35]. Consistent with previous observations, interestingly, our biochemical analysis results revealed that the serum levels of AST and ALT returned to near-normal levels after treatment with *E. faecium* strain 132 and *L. paracasei* strain 201, suggesting that the two strains were more likely to improve liver damage. In adipose and liver tissue, MCP-1 is a highly representative chemokine, which plays a key role in the pathogenesis of liver diseases, and it is the major determinant of monocyte/macrophage recruitment to the site of tissue injury [36]. IL-1β acts primarily as a pro-inflammatory mediator, activating and recruiting white blood cells, especially neutrophils, into tissues [37]. IL-6 is a cytokine involved in hematopoiesis, immune response and the regulation of acute and chronic inflammation [38], but it also promotes the production of pro-inflammatory C-reactive protein (CRP) by macrophages and T cells, which is associated with an increased risk of diabetes, hypertension and cardiovascular disease [39]. TNF-α is the key mediator of liver injury induced by LPS [40]. However, the current study showed that the HC-*E. faecium* strain 132 and HC-*L. paracasei* strain 201 groups significantly decreased the liver IL-1β and the epididymal adipose MCP-1 and IL-6 levels. The HC-*E. faecium* strain 132 group also had significantly decreased the liver MCP-1 and TNF-α. Histopathological examination revealed that hepatocyte steatosis and adipocyte hypertrophy in the HC*-L. paracasei* strain 201 and HC*-E. faecium* strain 132 groups were expectedly alleviated compared to those in the HC group. The results of this study have shown that *E. faecium* strain 132 and *L. paracasei* strain 201 are likely to improve liver injury and obesity by regulating liver and epididymal adipose inflammatory cytokines. 

Human gastrointestinal microbiota are not only a critical barrier to invasive substances, but they also regulate symbiotic homeostasis and normal physiological processes [41]. One study analyzed the correlation between the intestinal microbiota and abnormalities in 893 subjects and found that 34 species were significantly correlated with blood lipid levels, among which 6% were correlated with TC levels and 4% were correlated with HDL-C levels [42]. High-throughput sequencing of the 16S rRNA gene was performed using DNA extracted from feces samples for further microbiome analyses. From the alpha diversity index, treatment with lovastatin, *L. paracasei* strain 201 and *E. faecium* strain 132 could significantly improve the uniformity and complexity of the microbiota. From the beta diversity index, each group could be better separated by an unweighted UniFrac distance. In terms of the abundances of families, after intervention with lovastatin, *L. paracasei* strain 201 or *E. faecium* strain 132, those of Veillonellaceae and Erysipelotrichaceae and were significantly decreased compared to those in the HC group. The proportion of Veillonellaceae is positively correlated with all metabolic diseases caused by a high-fat diet [43]. In addition, studies have found that *Erysipelotrichaceae* can also cause abnormal lipid metabolism in the host [44]. At the genus level, the abundance of Lachnospiraceae and *Butyricicoccus* were significantly increased by treatment with *L. paracasei* strain 201 or *E. faecium* strain 132 compared to the HC group. The abundance of *Prevotella* was significantly decreased in the HC-lovastatin and HC-*E. faecium* strain 132 groups compared with that in the HC group. In studies on intestinal microbial transplantation, the abundance of *Prevotella* was positively correlated with plasma trimethylamine oxide (TMAO) level, indicating the influence of microorganisms on the formation of atherosclerotic lesions [41]. SCFAs not only serve as energy substrates, but also promote nutrient absorption, maintain microbial interactions and reduce the risk of cardiovascular events [45]. Studies have shown that both Lachnospiraceae and *Butyricicoccus* can produce short chain fatty acids (SCFAs), and butyrate and propionate have a protective effect on diet-induced obesity; propionate can especially reduce the synthesis of cholesterol in the liver and improve lipid metabolism [46,47,48]. The results have indicated that the *E. faecium* strain 132 or *L. paracasei* strain 201 might improve the microbiota to regulate the lipid-related metabolic balance. The inadequacy of our study is that we should have combined multiomics techniques to explore key substances and specific pathways.

## 5. Conclusions

In summary, oral administration of *E. faecium* strain 132 and *L. paracasei* strain 201, with their cholesterol-lowering excellent activity, ameliorated lipid metabolism, liver injury and gut microbiota. In terms of the cholesterol-lowering mechanisms, we found that genes were involved in cholesterol and lipid levels. *CYP8B1*, *CYP7A1, SREBP-1*, *SCD1* and *LDL-R* were regulated in hypercholesteremia rats intervened by two strains. However, they might also alleviate liver injury by reducing inflammation and ultimately increasing the production of acetic acid and propionic acid in fecal samples to improve hyperlipidemia in rats. These findings support that *E. faecium* strain 132 and *L. paracasei* strain 201 could have promising applications for the treatment of hypercholesteremia.

## Figures and Tables

**Figure 1 nutrients-13-01982-f001:**
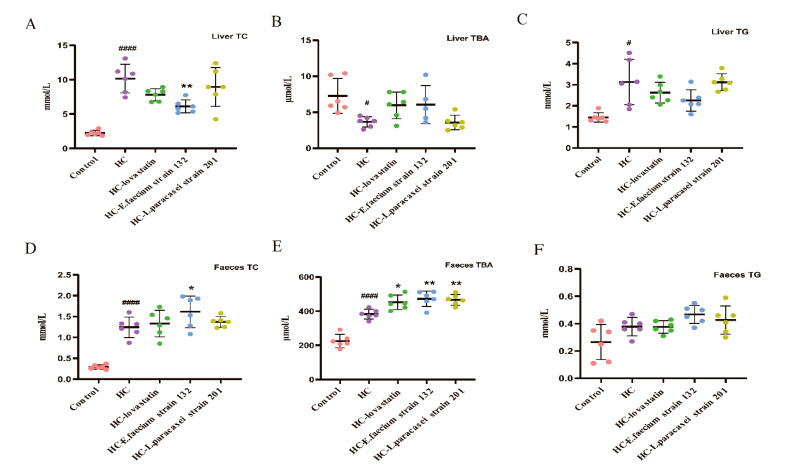
Levels of total bile acids (TBA), triglycerides (TG) and total cholesterol (TC) in feces and liver. (**A**) TC level in liver. (**B**) TBA level in liver. (**C**) TG level in liver. (**D**) TC level in feces. (**E**) TBA level in feces. (**F**) TG level in feces. Data are expressed as the mean ± SD of triplicate tests. (*n* = 6 for weight gained, TC, TG and TBA in feces and liver). The pink bar represents the control group, the purple bar represents the HC group, the green bar represents the HC-lovastatin group, the blue bar represents the HC-*E. faecium* strain 132 group and the yellowish-brown bar represents the HC-*L. paracasei* strain 201group. *p*-values were determined using a one-way ANOVA with Tukey’s test for post-hoc analysis. Significant differences between the HC group versus control group are indicated as *^#^ p* < 0.05, *^####^ p <* 0.0001. Significant differences in the HC-lovastatin, HC-*E. faecium* strain 132 or HC-*L. paracasei* strain 201 group versus HC group are indicated as ** p* < 0.05, *** p <* 0.01.

**Figure 2 nutrients-13-01982-f002:**
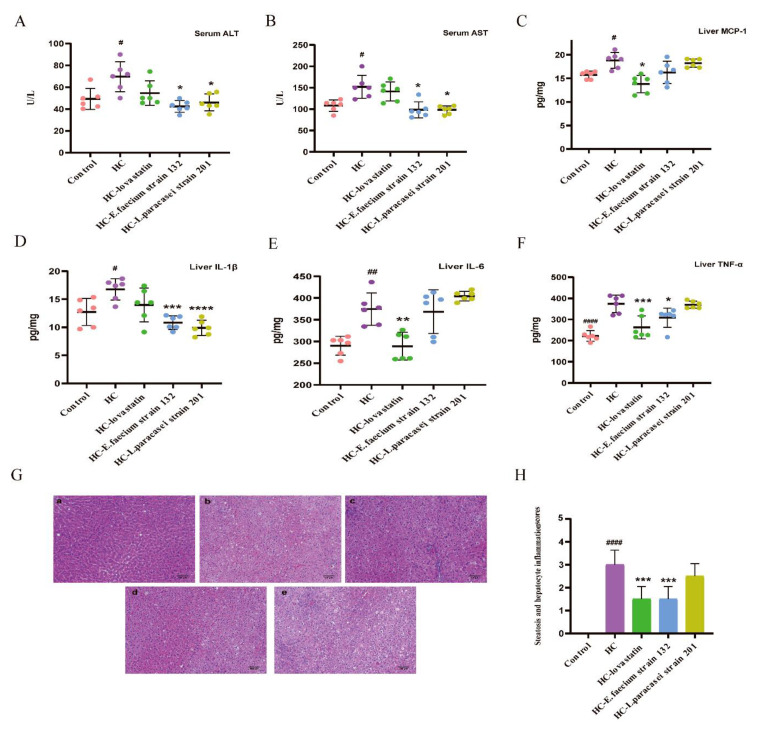
Screening strains improved liver injury in hypercholesterolemia rats. (**A**) The level of serum glutamic pyruvic transaminase (ALT). (**B**) The level of serum glutamic oxalacetic transaminase (AST). (**C**) The level of liver monocyte chemotactic protein-1 (MCP-1). (**D**) The level of liver interleukin-1β (IL-1β). (**E**) The level of liver interleukin-6 (IL-6). (**F**) The level of liver tumor necrosis factor-α (TNF-α). (**G**) Hematoxylin-eosin (H&E) staining of liver; (a) control group, (b) HC group, (c) HC-lovastatin group, (d) HC-*E. faecium strain* 132 group, (e) HC-*L. paracasei* strain 201 group. (**H**) Steatosis and hepatocyte inflammation scores. *p*-values were determined using a one-way ANOVA with Tukey’s test for post-hoc analysis, *n* = 6. Significant differences between the HC group versus control group are indicated as *^#^ p* < 0.05, *^##^ p* < 0.01, *^####^ p <* 0.0001. Significant differences in the HC-lovastatin, HC-*E. faecium* strain 132 or HC-*L. paracasei* strain 201 group versus HC group are indicated as ** p* < 0.05, *** p* < 0.01, **** p* < 0.001, ***** p <* 0.0001.

**Figure 3 nutrients-13-01982-f003:**
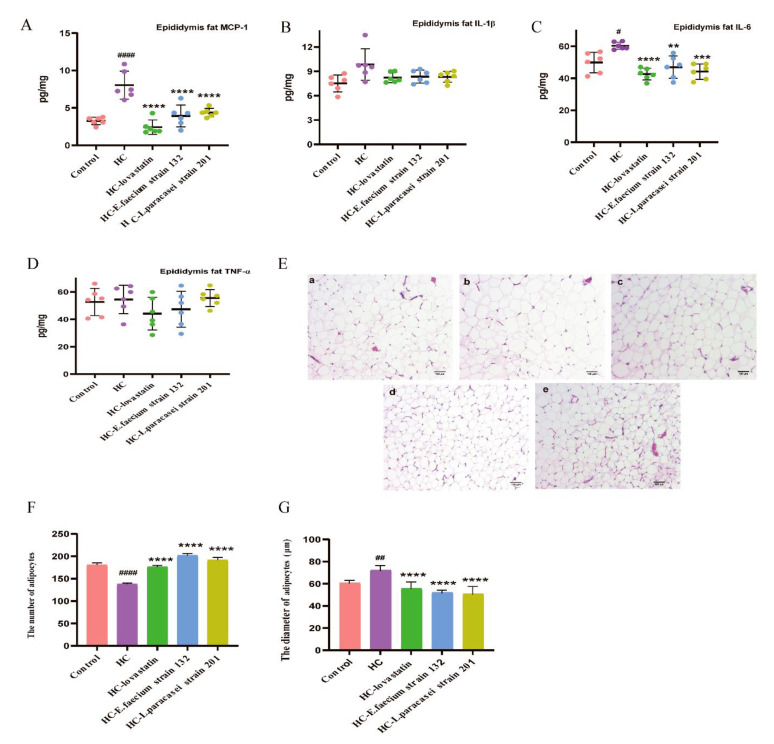
Screening strains improved the inflammation of epididymis fat and fatty hypertrophy in hypercholesterolemia rats. (**A**) The level of epididymis fat MCP-1. (**B**) The level of epididymis fat IL-1β. (**C**) The level of epididymis fat IL-6. (**D**) The level of epididymis fat TNF-α. (**E**) H&E staining of epididymis adipose; (a) control group, (b) HC group, (c) HC-lovastatin group, (d) HC-*E. faecium* strain 132 group, (e) HC-*L. paracasei* strain 201 group. (**F**) The number of adipocytes in the same field, (**G**) the diameter of adipocytes in the same field. *p*-values were determined using a one-way ANOVA with Tukey’s test for post-hoc analysis, *n* = 6. Significant differences between the HC group versus control group are indicated as *^#^ p* < 0.05, *^##^ p* < 0.01, *^####^ p <* 0.0001. Significant differences in the HC-lovastatin, HC-*E. faecium* strain 132 or HC-*L. paracasei* strain 201 group versus HC group are indicated as *** p* < 0.01, **** p* < 0.001, ***** p <* 0.0001.

**Figure 4 nutrients-13-01982-f004:**
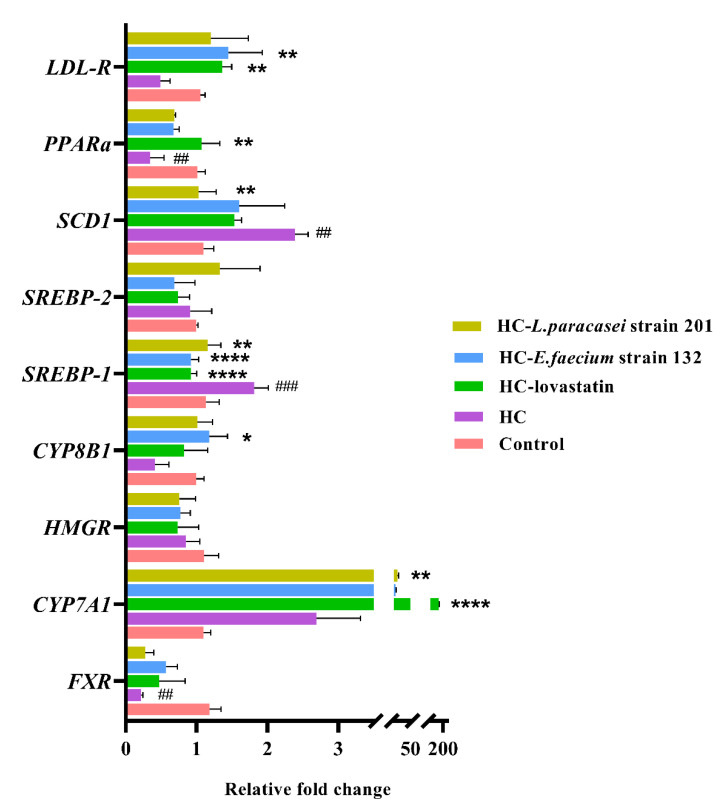
The mRNA expression levels of genes related to lipid metabolism in the liver. FXR stands for Farnesol X Receptor, and HMGR is named for 3-hydroxy-3-methyl glutaryl coenzyme A reductase. *p*-values were determined using a one-way ANOVA with Tukey’s test for post-hoc analysis, *n* = 6. Significant differences between the HC group versus control group are indicated as *^#^ p* < 0.05, *^##^ p* < 0.01, *^###^ p <* 0.001. Significant differences in the HC-lovastatin, HC-*E. faecium* strain 132 or HC-*L. paracasei* strain 201 group versus HC group are indicated as ** p* < 0.05, *** p* < 0.01, ***** p <* 0.0001.

**Figure 5 nutrients-13-01982-f005:**
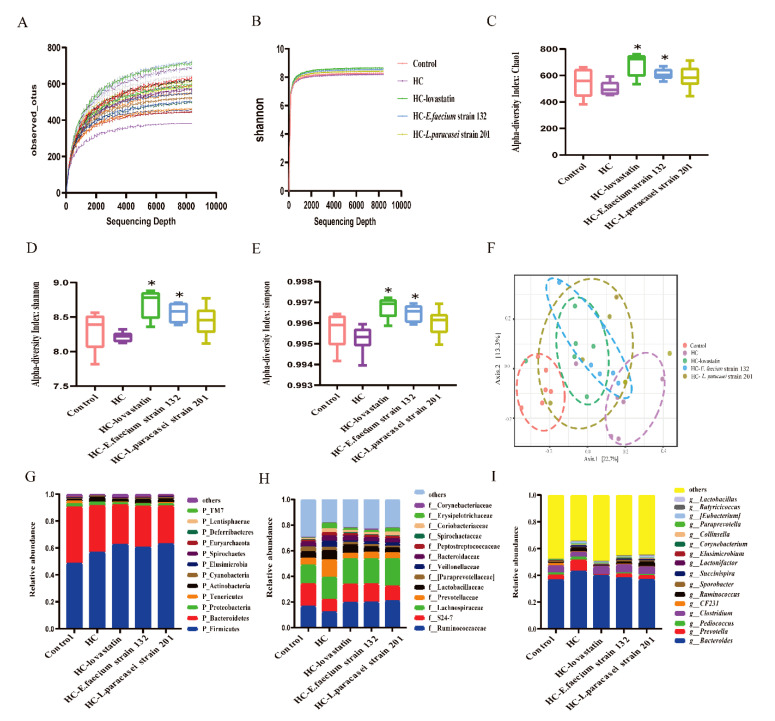
Rarefaction curve, alpha and beta diversity and the species abundance of phylum, family and genus levels. (**A**) Rarefaction curve based on observed_otus per sample (**B**) Rarefaction curve based on Shannon index per group. (**C**) Box plot of alpha diversity calculated by the Chao1 index. (**D**) Box plot of alpha diversity calculated by the Shannon index. (**E**) Box plot of alpha diversity calculated by the Simpson index. (**F**) Beta diversity calculated by Unweighted UniFrac Distance. (**G**) The relative abundance of phylum level. (**H**) The relative abundance of family level. (**I**) The relative abundance of genus level. The pink bar represents the control group, the purple bar represents the HC group, the green bar represents the HC-lovastatin group, the blue bar represents the HC-*E. faecium* strain 132 group and the yellowish-brown bar represents HC-*L. paracasei* strain 201group. A Kruskal–Wallis test was used to analyze the alpha diversity index, and the PERMANOVA test was used for beta diversity index analysis, *n* = 6. Significant differences in the HC-lovastatin, HC-*E. faecium* strain 132 or HC-*L. paracasei* strain 201 group versus HC group are indicated as ** p* < 0.05.

**Figure 6 nutrients-13-01982-f006:**
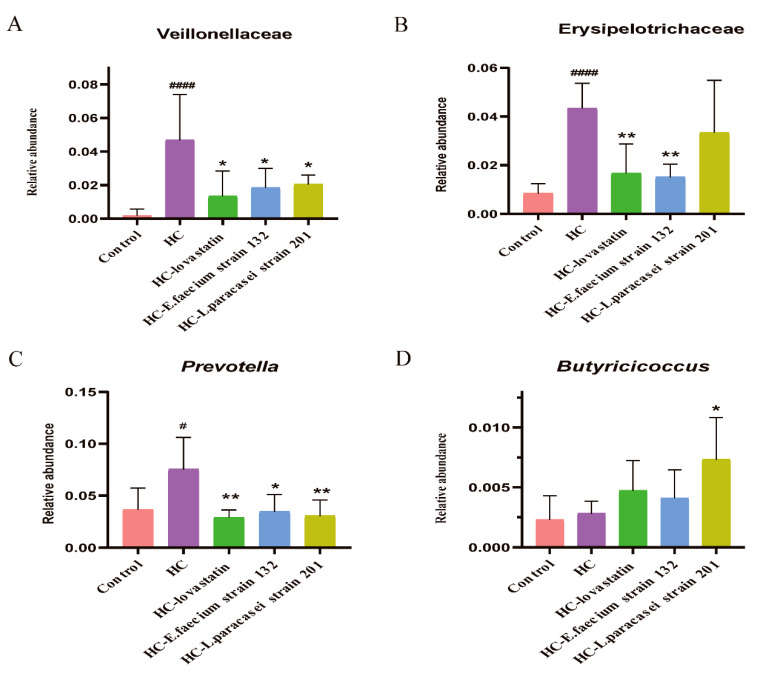
The relative abundance of Veillonellaceae, Erysipelotrichaceae, *Prevotella* and *Butyricicoccus*. (**A**) The relative abundance of Veillonellaceae; (**B**) The abundance of Erysipelotrichaceae; (**C**) The relative abundance of *Prevotella*; (**D**) The relative abundance of *Butyricicoccus*. *p*-values were determined using a one-way ANOVA with Tukey’s test for post-hoc analysis, *n* = 6. Significant differences between the HC group versus control group are indicated as *^#^ p* < 0.05, *^####^ p <* 0.0001. Significant differences in the HC-lovastatin, HC-*E. faecium* strain 132 or HC-*L. paracasei* strain 201 group versus HC group are indicated as ** p* < 0.05, *** p <* 0.01.

**Figure 7 nutrients-13-01982-f007:**
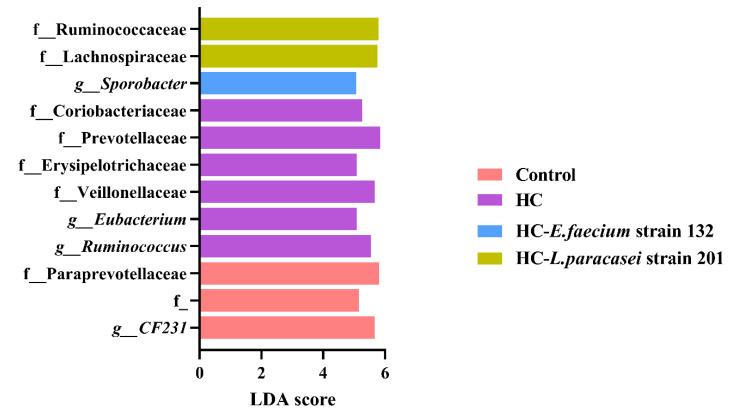
Linear discriminant analysis (LDA) and effect quantity (LEfSe) were used to compare the microflora distribution among groups. LDA score > 4, FDR value < 0.1. The pink bar represents the control group, the purple bar represents the HC group, the blue bar represents the HC-*E. faecium* strain 132 group and the yellowish-brown bar represents the HC-*L. paracasei* strain 201group.

**Table 1 nutrients-13-01982-t001:** Primer sequences of related genes.

Gene	Oligonucleotide Sequence (5′-3′)
*CYP7A1* F	GCCTTCCTATTCACTTGTTC
*CYP7A1* R	GTGGAGAGCGTGTCATTG
*CYP8B1* F	CCTCTTCCACTTCTGCTAC
*CYP8B1* R	GTCCTGCTCCTTGTCCTT
*HMGR* F	GGTGCGAAGTTCCTTAGTGAT
*HMGR* R	ATGAGGGTTTCCAGTTTGTAGG
*SREBP-2* F	CAGCAGCAGACGGTGATGA
*SREBP-2* R	TGGTTGCGGCATTCTGGTAT
*SREBP-1* F	ACACAGACAAACTGCCCATC
*SREBP-1* R	TCATTGATAGAGGAACGGTAGC
*PPARα* F	CTTCACGATGCTGTCCTCCT
*PPARα* R	GATGTCGCAGAATGGCTTCC
*LDL_R* F	TGTGGCAGTAGTGAGTGT
*LDL_R* R	GTTCTCCTCGTCCGACTT
*FXR* F	TCCTCCTCGTCCTATTATTCCA
*FXR* R	GCATTCGCCTGAGTTCATAGA
*SCD1* F	TCCTACACGACCACCACTAC
*SCD1* R	GGCACCTTCTTCATCTTCTCT

**Table 2 nutrients-13-01982-t002:** The total weight gain, organ weight and blood biochemical index of rats in each group.

Parameters	Groups				
	Control	HC	HC-Lovastatin	HC-*E. faecium* Strain 132	HC-*L. paracasei* Strain 201
Weigh gained (g/6 weeks)	155.02 ± 32.60	184.31 ± 26.32 ^###^	147.03 ± 30.68 ***	156.91 ± 20.57 **	165.14 ± 22.65 **
Cardiac index	0.29 ± 0.03	0.30 ± 0.03	0.29 ± 0.04	0.31 ± 0.03	0.30 ± 0.02
Liver index	2.50 ± 0.35	3.79 ± 0.60 ^###^	3.11 ± 0.25 **	3.10 ± 0.51 *	2.95 ± 0.46 **
Spleen index	0.19 ± 0.02	0.21 ± 0.09	0.18 ± 0.04	0.19 ± 0.04	0.22 ± 0.09
Kidney index	0.60 ± 0.06	0.57 ± 0.07	0.54 ± 0.08	0.56 ± 0.05	0.55 ± 0.08
Epididymal fat index	0.99 ± 0.18	1.41 ± 0.29 ^##^	0.98 ± 0.19 *	1.08 ± 0.12 *	0.99 ± 0.14 *
TBA(μmol/L)	13.98 ± 5.81	14.52 ± 5.99	13.51 ± 7.78	13.97 ± 3.92	14.47 ± 8.15
TG (mmol/L)	0.33 ± 0.11	0.63 ± 0.12 ^###^	0.24 ± 0.09 ****	0.38 ± 0.08 **	0.40 ± 0.04 **
LDL-C(mmol/L)	0.27 ± 0.04	0.93 ± 0.06 ^####^	0.67 ± 0.22 **	0.71 ± 0.11 *	0.69 ± 0.05 *
HDL-C(mmol/L)	0.46 ± 0.19	0.26 ± 0.04 ^##^	0.25 ± 0.07	0.29 ± 0.05	0.28 ± 0.05
TC (mmol/L)	1.42 ± 0.23	1.91 ± 0.70 ^#^	1.63 ± 0.53	1.71 ± 0.15	1.69 ± 0.31

Organs index = (organ weight/total weight) ×100; *p*-values were determined using a one-way ANOVA with Tukey’s test for post-hoc analysis, *n* = 6. Significant differences between the HC group versus control group are indicated as *^#^ p* < 0.05, *^##^ p* < 0.01, *^###^ p* < 0.01, *^####^ p <* 0.0001. Significant differences between the HC-lovastatin, HC-*E. faecium* strain 132 or HC-*L. paracasei* strain 201 group versus from HC group are indicated as ** p* < 0.05, *** p* < 0.01, **** p* < 0.001, ***** p <* 0.0001. TBA, total bile acid; TG, triglycerides; LDL-C, low-density lipoprotein cholesterol; HDL-C, high-density lipoprotein cholesterol; TC, total cholesterol.

**Table 3 nutrients-13-01982-t003:** The content of short chain fatty acids.

Short Chain Fatty Acid	Control	HC	HC-Lovastatin	HC-*E. faecium* Strain 132	HC-*L. paracasei* Strain 201
Acetic acid	3.01 ± 0.36	2.42 ± 0.81	3.42 ± 0.04 *	3.60 ± 0.12 *	3.30 ± 0.19
Propionic acid	0.95 ± 0.10	0.81 ± 0.28	1.21 ± 0.05 *	1.38 ± 0.12 **	1.17 ± 0.05 *
Isobutyric acid	0.32 ± 0.01	0.37 ± 0.14	0.31 ± 0.01	0.31 ± 0.01	0.31 ± 0.00
Butyric acid	0.64 ± 0.02	0.57 ± 0.16	0.57 ± 0.01	0.60 ± 0.02	0.60 ± 0.02
Isovaleric acid	0.33 ± 0.00	0.33 ± 0.01	0.33 ± 0.00	0.32 ± 0.01	0.32 ± 0.00
Valeric acid	0.49 ± 0.00	0.49 ± 0.06	0.48 ± 0.00	0.48 ± 0.01	0.49 ± 0.00

*p*-values were determined using a one-way ANOVA with Tukey’s test for post-hoc analysis, *n* = 6. Significant differences in the HC-lovastatin, HC-*E. faecium* strain 132 or HC-*L. paracasei* strain 201 group versus HC group are indicated as ** p* < 0.05, *** p* < 0.01.

## Data Availability

16S amplicon data from animal feces of SD rats http://www.ncbi.nlm.nih.gov/bioproject/698075 (accessed on 17 May 2021) Submission ID: SUB8969130, BioProject ID: PRJNA698075, 16s rRNA gene sequence of *Enterococcus faecium* strain 132, *Lactobacillus paracasei* 201, *Enterococcus faecium* strain 132: SUB9198052 _A13_1912250513Q.seq.Contig1 MW692152, *Lactobacillus paracasei* strain 201: SUB9198052 _A23_1912250523Q.seq.Contig1 MW692153.

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
