# Peer review of "Evaluation of the Cholesterol-Lowering Mechanism of *Enterococcus faecium* Strain 132 and *Lactobacillus paracasei* Strain 201 in Hypercholesterolemia Rats"

_nutrients, 2021, doi:10.3390/nu13061982_

Round 1

Reviewer 1 Report

The topic of the article falls within the thematic scope of the journal.

The aim of this study was to assess 2 human faecal isolates (Lb. paracasei and E. faecium) for their cholesterol-lowering capacity in vivo (in rats) and in vitro; explore the specific mechanism through which these strains protect against liver injury, epididymal fat, inflammation, lipid metabolism, and variations in the intestinal microbiota associated with hypercholesterolaemia in rats.

The subject is very interesting, the scope of the research was very wide, and the manuscript is well prepared.

I have no objections to the Abstract, Introduction and Results, and their discussions;

but I have one to section Materials and methods: There is no description of what happened next with the strains (desription in 2.1); it can only be guessed that two strains were selected: Lb. paracasei 201 and E. faecium 132 and these were further investigated. It is necessity to complete this information.

All minor remarks (lines 28, 30, 37, 49, 90, 406, 412, 413, 420428) are marked in the text of manuscript in the review mode.

After introducing these minor changes, in my opinion, the Editors may direct the manuscript for further processing.

Author Response

Point 1: I have one to section Materials and methods: There is no description of what happened next with the strains (description in 2.1); it can only be guessed that two strains were selected: Lb. paracasei 201 and E. faecium 132 and these were further investigated. It is necessity to complete this information.

Response 1: Thank you for your advice, we have added the description here (lines 109-112). We have also added the description in 2.2 (lines 131-133).

Point 2: All minor remarks (lines 28, 30, 37, 49, 90, 406, 412, 413, 420, 428) are marked in the text of manuscript in the review mode.

Response 2: Thank you for your review and advice, we have checked and corrected the relevant errors and marked them in yellow (lines 28, 30, 37, 49, 94, 442, 448, 449, 455, 463).

Reviewer 2 Report

The subject of the article very interesting, once Hypercholesterolemia is a major risk factor of cardiovascular disease and E. faecium strain 132 and L. paracasei strain 201 could be a promising application for the treatment of hypercholesteremia.  

General comments:

  1. The introduction is very good
  2. In the figure the author say “Control group are indicated as #P < 0.05, 0.0001. However, after the indication appears in the control bar, shouldn't it appear in the one you are buying with the control ? In addition, there is no purchase between control and the other HC groups? In my opinion there must be an error in the placement of the statistics in the graphs.
  3. Add the statistical test used in the figure legend.
  4. In the figure 5 G, H the caption are no visible, pleased improve image quality.
  5. The description in table 3 is not complete.
  6. The conclusion is good

Author Response

Point 1: In the figure the author say “Control group are indicated as #P < 0.05, 0.0001. However, after the indication appears in the control bar, shouldn't it appear in the one you are buying with the control? In addition, there is no purchase between control and the other HC groups? In my opinion there must be an error in the placement of the statistics in the graphs.

Response 1: Thank you very much for your suggestion, we have changed the placement of the statistics in the graphs (Table 2 and Figure 1,2,3,4,6; lines 247, 264, 296, 316, 342, 410).

Point 2: Add the statistical test used in the figure legend

Response 2: Thank you very much for your review and suggestion, we have added the statistical test used in the figure legend (Figure 1,2,3,4,6; lines 270-274, 301-302, 323-324, 343-344, 411-412).

Point 3: In the figure 5 G, H the caption are no visible, pleased improve image quality.

Response 3: Thank you very much for your suggestion, we check and change the quality of image, and the resolution reach 1200 dpi (figure 5 G, H, line 364).

Point 4: The description in table 3 is not complete.

Response 4: Thank you very much for your comment and suggestion, we have added the description of table 3 (lines 434-436).

Round 2

Reviewer 2 Report

The authors have revised the manuscript according to the reviewer's comments.

I recommend accept

Author Response

Thank you